# Dysfunctional Lipid Metabolism—The Basis for How Genetic Abnormalities Express the Phenotype of Aggressive Prostate Cancer

**DOI:** 10.3390/cancers15020341

**Published:** 2023-01-04

**Authors:** Matthew Alberto, Arthur Yim, Nathan Lawrentschuk, Damien Bolton

**Affiliations:** 1Department of Urology, Austin Health, University of Melbourne, Melbourne, VIC 3010, Australia; 2Department of Urology, Royal Melbourne Hospital, Melbourne, VIC 3010, Australia

**Keywords:** prostate cancer, lipid metabolism, genetics, androgen deprivation

## Abstract

**Simple Summary:**

Advanced prostate cancer has a higher mortality rate at diagnosis compared to localised prostate cancer. As such, it is critical to understand the mechanisms of development, and potential pathways that may drive research into novel treatments. We aim to review how lipid metabolism relates to advanced prostate cancer.

**Abstract:**

Prostate cancer is the second most frequent cancer in men, with increasing prevalence due to an ageing population. Advanced prostate cancer is diagnosed in up to 20% of patients, and, therefore, it is important to understand evolving mechanisms of progression. Significant morbidity and mortality can occur in advanced prostate cancer where treatment options are intrinsically related to lipid metabolism. Dysfunctional lipid metabolism has long been known to have a relationship to prostate cancer development; however, only recently have studies attempted to elucidate the exact mechanism relating genetic abnormalities and lipid metabolic pathways. Contemporary research has established the pathways leading to prostate cancer development, including dysregulated lipid metabolism-associated de novo lipogenesis through steroid hormone biogenesis and β-oxidation of fatty acids. These pathways, in relation to treatment, have formed potential novel targets for management of advanced prostate cancer via androgen deprivation. We review basic lipid metabolism pathways and their relation to hypogonadism, and further explore prostate cancer development with a cellular emphasis.

## 1. Introduction

Prostate cancer (PCa) is the second most frequent cancer and the fifth most common cause of cancer death in men according to the most recent GLOBOCAN data [1]. In 2020 alone an estimated 1,414,000 new cases and 375,304 deaths occurred globally [2]. Fundamental changes in genetics and lipid metabolism drive the growth of PCa cells, leading to development and progression of disease. The mechanisms behind such growth are important to understand given the increasing burden of disease secondary to an ageing population, further emphasizing the need to continually develop treatment strategies. We aim to provide a comprehensive review of historical and contemporary evidence for dysregulated lipid metabolism, its relationship to hypogonadism, PCa cellular pathways and genetic abnormalities.

## 2. General Lipid Metabolism

Lipid metabolism involves a balance of synthesis and degradation of structural and functional lipids to satisfy the metabolic needs of the body and maintain dynamic equilibrium [3]. Examples of lipids include fatty acids (FA), phospholipids, glycolipids, cholesterol and prostaglandins. Relevant mechanisms of lipid metabolism for this review involve biosynthesis of steroid hormones and β-oxidation of FA, both increasingly studied in PCa development and progression [4]. Each pathway will be discussed, with relevance to PCa examined in more detail later.

Cholesterol is important in the biosynthesis of steroid hormones, of which testosterone (androgen) and its metabolite dihydrotestosterone (DHT) are classified as sex-steroids. All steroid hormones are generated from cholesterol via a common pathway involving pregnenolone, a steroid precursor. Cholesterol is converted to pregnenolone via cytochrome P450 cholesterol side-chain cleavage (P450scc) enzyme (CYP11A1) [5]. Subsequent formation of dehydroepiandrosterone (DHEA) and androstenedione occurs in the adrenocortical cells of the zona fasciculata and zona reticularis layers, respectively, allowing testosterone formation in testicular Leydig cells [5]. Centralised control of testosterone production occurs in the hypothalamus. The hypothalamus initiates pulsatile release of luteinising hormone-releasing hormone (LHRH) which binds and stimulates LHRH receptors in the anterior pituitary gland causing release of luteinising (LH) and follicle-stimulating hormones (FSH). Leydig cells in the testis are stimulated by LH and induce production of testosterone and subsequent conversion to its more potent form DHT via 5 α-reductase [5]. 

FA metabolism is considered critical in PCa, influencing several pathways including cell signalling, energy processing and membrane fluidity [6]. Its importance in cancer is elevated by increased energy demands to propagate growth and progression [7]. However, β-oxidation of FAs is considered the most prominent oxidation reaction as it relates to peroxisomal (membrane-bound organelle important in oxidative reactions) β-oxidation. This is required for the initial oxidation of very long chain FAs, branched chain FAs and derivatives not able to be directly oxidised by the mitochondrion [6]. In normal cells, FAs are oxidised to acetyl-coenzyme A (CoA) via multiple pathways in β-oxidation methods in peroxisomes and mitochondria. Peroxisomal β-oxidation can occur via branched chain acyl-CoA oxidase (ACOX2) and/or pristanoyl-CoA oxidase (ACOX3) and via D-bifunctional protein (DBP) [8]. Furthermore, α-methylacyl-CoA racemase (AMACR) is an enzyme in both peroxisomes and mitochondria aiding β-oxidation of branched FAs. Subsequently, oxidation of FAs to acetyl-CoA allows initiation of the tricarboxylic acid cycle (TCA) for production of adenosine triphosphate (ATP), the principal energy substrate intrinsic to cellular function and division [6].

## 3. Lipid Metabolism, Hypogonadism and Testosterone Disorders—Chicken or the Egg?

It has long been known from epidemiological data that increased testosterone levels are associated with a favourable lipid profile, that is, a lower total cholesterol, LDL and triglycerides (TG) and a higher HDL [9]. Subsequent interventional trials of testosterone replacement have demonstrated its ability to improve lipid profiles, as it suppresses proinflammatory cytokines TNFα, IL-1β and IL-6 to reduce the inflammatory state and lower the total cholesterol profile [10]. Meanwhile, testosterone deficiency is associated with an increased risk of developing metabolic disorders and is also highly prevalent in obesity, metabolic syndrome (MetS) and type 2 diabetes. Models of gonadotrophin releasing hormone deficiency—and androgen deprivation therapy (ADT) in patients with PCa—suggest that hypogonadotropic hypogonadism contributes to the onset and worsening of metabolic conditions, by increasing visceral adiposity and insulin resistance [11]. Therefore, the relationship between lipid metabolism and hypogonadism is bidirectional, in the context of testosterone deficiency and metabolic disease, both of which may have a roll in each other’s pathogenesis.

Hypogonadism in men is characterised by an impairment in gonadal function resulting in circulating testosterone levels below the normal range. The clinical syndrome of testosterone deficiency includes impaired spermatogenesis, sexual dysfunction, reduction in testis volume and gynecomastia, together with anaemia, deterioration of muscle mass and metabolic abnormalities [12]. Hypogonadism can be classified as primary or secondary, as well as organic or functional (Figure 1). In primary hypogonadism, testicular failure is the underlying cause, resulting in increased gonadotropin levels and this is defined as hypergonadotropic hypogonadism (Hyper-T). In secondary hypogonadism, the central hypothalamus–pituitary function is disrupted, leading to hypogonadotropic hypogonadism (Hypo-H) [12]. Organic hypogonadism is caused by specific pathologies that are well-recognised, such as Klinefelter’s Syndrome (KS), Cushing syndrome, pituitary injury, prolactinoma or testicular trauma and is relatively rare. KS is a prime example of an organic, primary Hyper-T hypogonadism that is strongly associated with MetS. Observational studies in Korean KS patients have shown significantly worsened dyslipidaemia, especially elevated TGs and decreased HDL levels [13].

However, the absence of hypothalamus–pituitary–thyroid (HPT) axis pathology in the setting of hypogonadism-like features and lowered circulating testosterone are characteristic of functional hypogonadism [14]. Chronic diseases such as dyslipidaemia, diabetes, depression, renal or liver diseases and obesity are characteristic of men with functional hypogonadism. Furthermore, functional hypogonadism is demonstrated to have an association with ageing, thereby increasing prevalence. Men aged 50–59 years have 0.6% prevalence, rising to 5.1% in those aged 70–79 [15]. ‘Late-onset’ hypogonadism has been coined secondary to these observations; as such, these men have modest reductions in circulating testosterone concentrations, approximately 6–10 mmol/L, compared to reference young healthy men [16]. However, the extent to which lowered testosterone concentrations contribute to the ageing male phenotype is not known. Another theory is that functional hypogonadism may be secondary to an accumulation of age-related co-morbidities and MetS, instead of purely due to ageing [17]. HPT suppression may also simply occur due to poor health, leading to lowered testosterone [14]. As such, age-related testosterone reduction may be preventable through health and lifestyle management. Nevertheless, low testosterone is at the very least a sensitive marker of suboptimal health. 

**Figure 1 cancers-15-00341-f001:**
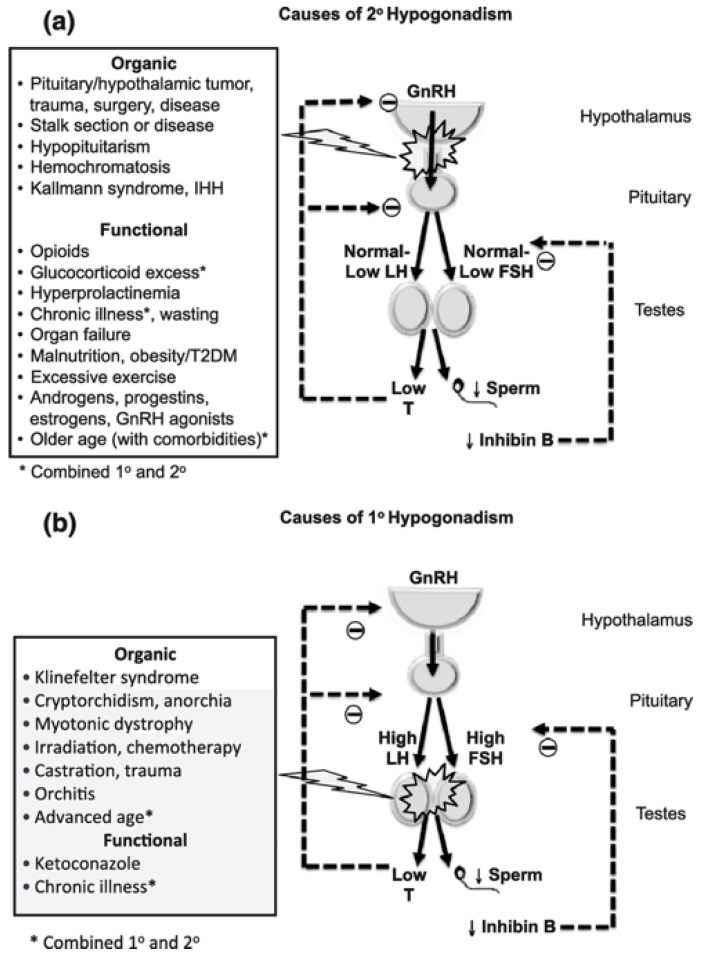
Classified causes of hypogonadism by Grossmann and Matsumoto via PMC Open Access Subset [16]. * is noted to be “combined primary and secondary hypogonadism”.

The exact pathophysiological mechanism by which testosterone deficiency leads to metabolic impairment, contributing to obesity, dyslipidaemia and type 2 diabetes, is still unclear. In a study on ageing hypercholesterolaemic men, clinically significant elevations of lipoprotein(a) were found in men with low testosterone [18]. It has therefore been proposed that low testosterone induces lipoprotein(a) lipase activity resulting in increased FA uptake and TG formation in adipocytes, which ultimately stimulate adipocyte proliferation and accumulation of adipose tissue, especially visceral adiposity [19]. On the other hand, mechanistic studies in hypogonadal men have suggested that increased adiposity leads to increased aromatisation of testosterone into oestradiol. As oestradiol increases, testosterone levels decrease, which can then result in a further unfavourable lipid profile [20]. This clearly demonstrates a vicious cycle involving hypogonadism, obesity and an unfavourable metabolic profile (Figure 2). This profile leads to dyslipidaemia, type 2 diabetes and increased visceral adiposity, thus further decreasing testosterone levels. In fact, men with PCa who receive ADT are prime examples of this phenomenon. As ADT is commenced and testosterone levels reduce, a detrimental effect is seen not only in the patient’s lipid profile, but a whole host of cardiovascular perturbations, including MetS, higher blood pressure, left ventricular hypertrophy and overall mortality [21].

## 4. Prostate Cancer and Dysregulated Lipid Metabolism

The relationship between lipid metabolism and cancer was first observed by Medes et al. [23] in 1953. Cancer tissue was found to overexpress enzymes to generate FAs and phospholipids via de novo lipogenesis in conjunction with environmental uptake of lipids. De novo lipogenesis, in turn, supports the excess energy requirements for growth and proliferation which has become a notable hallmark of cancer [24]. Cells can utilise FAs for energy generation via β-oxidation to generate ATP, the principal energy molecule. Activation of the de novo lipogenesis pathway affects all levels of lipid enzyme regulation, occurring downstream to known oncogenic abnormalities such as activation of akt, loss of PTEN, mutation or loss of p53 or BRCA1 and steroid hormone activation [4]. Additionally, exogenous lipids from circulation and lipolysis or stored lipids in adipocytes and intracellular lipid droplets can also be utilised [25].

Interestingly, PCa cells differ from various other cancers as FAs are the predominant energy substrate as opposed to glucose [26]. PCa cells undergo dysregulated lipid metabolism comprising increased de novo lipogenesis in the form of steroid hormone biosynthesis and β-oxidation of FAs for energy generation, membrane synthesis and cell division. Furthermore, FAs are stored in lipid droplets or converted to complex phospholipids as key components to cell membranes [4]. Remarkably, epidemiologic data suggests obesity is a significant risk factor for aggressive forms of PCa, further emphasising the role of dysregulated lipid metabolism [27]. Laboratory studies have demonstrated FA synthase (FAS) having similar properties to oncogenes in PCa mouse models and FAS inhibitors having converse effects, limiting PCa growth in similar environments [28,29]. As such, these studies further emphasise lipid metabolism as a significant contributor to PCa development, although theorised mechanisms will be explored in a later section.

## 5. Lipid Metabolism and Androgens in Prostate Cancer

Huggins and Hodges [30] first noted in 1941 the improvement of patients with metastatic PCa when chemically castrated with oestrogens. This led to the understanding that PCa is exquisitely influenced by androgenic activity with inhibition occurring with elimination of androgens. Androgen receptor (AR) activation is a key player in PCa growth and stimulation via multiple metabolic pathways, and its link to lipid metabolism has been observed in advanced PCa, whereby accumulation of lipid droplets in the cytoplasm occurs via AR-associated increased synthesis of cholesterol and FAs as demonstrated in Figure 3 [31,32]. Furthermore, an AR antagonist reverses the effects of lipogenesis, which is not seen in AR-negative PCa cells [32]. Subsequently androgens have been found to influence prostate cell lipid profile through synthesis, binding, uptake, metabolism and transport of lipids [4].

The most characterised mechanism for androgen involvement in stimulation of de novo lipogenesis is via indirect regulation of protein expression, a transcription factor family named sterol regulatory element-binding protein (SREBP). SREBP plays an important part in increasing lipid and cholesterol metabolism and, in turn, aids androgen synthesis [33]. Specifically, SREBPs are comprised of SREBP1a and SREBP1c, and with two isoforms are master regulators of lipid homeostasis due to regulation of enzymes required for lipid synthesis and uptake. Reduction of intracellular sterol levels causes SREBP cleavage activating protein (SCAP)-SREBP complex translocation into the Golgi and is further cleaved by proteases (site-1 and site-2). This, in turn, causes SREBP to translocate to the nucleus, binding to sterol-response elements (SRE) inducing transcription of the key enzymes to de novo lipogenesis, including FAS, 3-hydroxy-4-methylglutarul coenzyme A reductase (HMG CoA-R) and LDL receptor (LDLR) [34]. Furthermore, several studies reviewed by Wu et al. [35] have emphasised the importance of FAs as a dominant energy source in PCa, finding increased expression of enzymes DBP and AMACR, noted earlier to be important in β-oxidation for ATP generation [6]. Given the exquisite relationship of androgens for stimulation and growth of PCa, the mainstay of treatment of advanced PCa is ADT, which inhibits testicular testosterone production either medically or surgically to reduce circulating levels of androgen [36].

**Figure 3 cancers-15-00341-f003:**
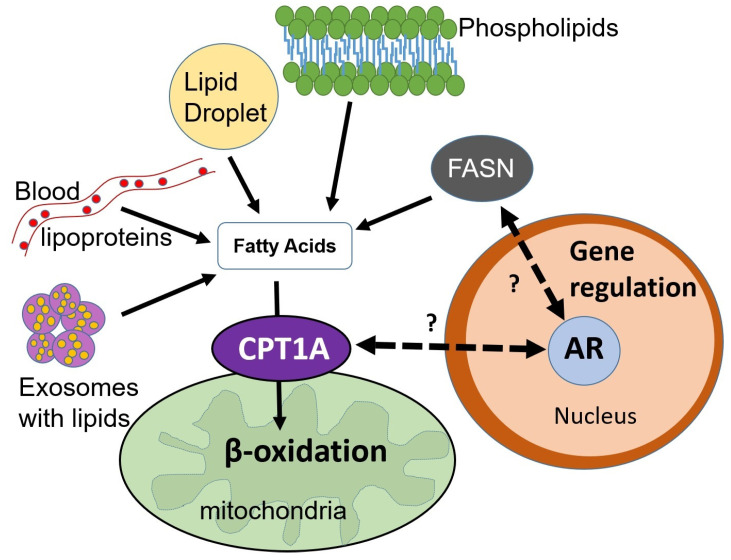
Androgen control of lipid metabolism by Stoykova and Schlaepfer via Creative Commons CC BY 4.0 [37].

## 6. Androgen Deprivation Therapy

Despite the availabilities of newer targeting agents, such as Enzalutamide or Abiraterone, classic ADT (hormonal therapy) has widespread use for local advanced to metastatic hormone sensitive PCa as neoadjuvant or adjuvant therapy with radiotherapy. Furthermore, addition of ADT to other systemic agents, such as AR targeted therapy, have recently been noted to improve overall survival in a systematic review when comparing systemic treatments for metastatic castration-sensitive (hormone sensitive) PCa [38]. Wang et al. [38] found Abiraterone acetate (hazard ratio (HR), 0.61; 95% confidence interval (CI), 0.54–0.70) and Apalutemide (HR, 0.67; 95% CI, 0.51–0.89) may have the most improvement to overall survival when added to ADT. A historic form of ADT, which remains a treatment option, is bilateral orchidectomy. Though superseded by more commonly used non-surgical treatments, there remain benefits with surgical castration including less cost and follow-up with potentially fewer side effects. Furthermore, Weiner et al. [39] found that survival rates are comparable between surgical and medical castration.

ADT targets various portions of the hypothalamic–pituitary–gonadal axis and are broadly classed into antiandrogens, LHRH agonist/antagonists (i.e., gonadotrophin releasing hormone (GhRH)) and androgen pathway inhibitors [40]. Antiandrogens block the AR to reduce testosterone cellular signalling whilst androgen pathway inhibitors target along the androgen pathway to reduce AR signalling or inhibit testosterone synthesis. LHRH agonists/antagonists target the LHRH receptor in the anterior pituitary gland. LHRH agonists stimulate the receptor leading to a temporary LH and testosterone surge, subsequently downregulating the receptor causing reduction in the LH and testicular production. Conversely, LHRH antagonists are competitive reversible agents blocking the LHRH receptors, in turn reducing LH release and therefore dropping testosterone production which avoids the initial transient rise. The first systematic review and meta-analysis of prospective studies in men undergoing ADT and its effects on body composition in PCa demonstrated significant increase in body fat on average of 7.7% (95% CI, 4.3–11.2, *p* < 0.0001), body weight (2.1%, *p* < 0.001) and BMI (2.2%, *p* < 0.001) with reduction in lean body mass −2.8% (95% CI, −3.6, −2.0, *p* < 0.0001) [41]. This data was corroborated by a more recent systematic review of 39 studies, which similarly found increased body fat mass and body weight with decrease in lean mass [42]. Indeed, this emphasises the effect of ADT on lipid metabolism and the role of androgen in PCa stimulation and growth as previously described. Authors also note that given the side effects, ADT may increase the risk of several other co-morbid conditions associated with MetS substantiated by recent reviews [43,44]. ADT causes suppression of circulating androgens with hypogonadism in some cases within 2–3 days of a loading dose, such as in LHRH antagonists (e.g., Degarelix) [45].

## 7. Cellular Mechanisms of Prostate Cancer and Dysfunctional Lipid Metabolism

Several mechanisms for dysregulated lipid metabolism in PCa via genetic abnormalities have been hypothesised, and this section will mainly focus on two novel theories. The first focuses on amplification and overexpression of pyruvate dehydrogenase complex (PDC), which is a gatekeeper for conversion of pyruvate to acetyl-CoA and subsequent entry into the mitochondrial TCA cycle [46]. Another focuses on co-deletion of promyelocytic leukemia (PML) and phosphatase and tensin homolog (PTEN) on PTEN-null PCa phenotypes [47]. PTEN is a commonly mutated or lost tumour-suppressor gene in many cancers [48] with partial loss in up to 70% of localised PCa [49] and complete loss linked to metastatic castration-resistant PCa [50]. The PTEN-null transgenic PCa mouse model was utilised in both these studies and emulates high grade intraepithelial prostate tumours at an early age and invasive PCa at late age.

Chen et al. [46] emphasise the association of increased mitochondrial metabolism and cancer pathogenesis and progression through the investigation of PDC. A major component of PDC is PDHA1, which is activated when dephosphorylated by pyruvate dehydrogenase phosphatase (PDP). Conversely, PDC is deactivated when phosphorylated by pyruvate dehydrogenase kinases (Pdks). The authors first establish that subunits of the PDC are amplified and overexpressed in the PTEN-null transgenic mouse model similarly to clinical PCa. In vivo analysis of both mouse and human PCa models demonstrated hampering of PCa progression via suppression of lipid biosynthesis when PDHA1 is inactivated [46]. Furthermore, the authors postulate that the PDC must be functional in both mitochondria and cancer cell nuclei for sufficient support of lipid biosynthesis. In the nucleus, SREBP transcription factor, previously described as important in de novo lipogenesis, has reduced histone acetylation at regulatory regions. However, at the mitochondrial level lipid biosynthesis suppression was noted to be due to a reduction in citrate production, which is important for ATP generation via the TCA cycle [46]. Finally, the authors postulate that PDC, specifically PDHA1, may be a potential therapeutic target for prevention of PCa development. A recent study similarly suggested the therapeutic potential for targeting PDC components and comparatively also found a positive correlation between PDC component (PDHA1, PDP1 and PDK) expression and AR expression [51].

Alternatively, Chen et al. [47] aimed to explore the impact of PML and PTEN co-deletion in metastatic PCa in PTEN-null transgenic mouse of PCa. Amplification of lipid metabolism was identified on coordinated loss of these two tumour suppressor genes. Similar to the PDC study [46], this study identifies amplification of an SREBP-dependent lipogenic program, albeit through hyperactivation of MAPK signalling, which is known to be limited by PML [47]. Interestingly, inhibition of SREBP by fatostatin can block metastasis, which can be replicated by a high fat diet in PTEN-null mice without PML loss [47]. Therefore, the authors note that PML deletion leads to amplification of MAPK signalling and subsequent aberrant lipid metabolism. Current evidence for statin use in PCa has no consensus guidelines; however, a recent systematic review and meta-analysis suggests statins may have a unique role in the reduction of biochemical recurrence of PCa post definitive treatment [52]. 

Several other genetic drivers of aggressive PCa include p53, retinoblastoma (RB1) and Myc, which are well established mechanisms of cancer cell survival in nutrient-poor environments. P53, RB1 and Myc may also be associated with dysfunctional lipid metabolism. Tumour suppressor gene (TSG) p53 regulates cellular metabolism and is a transcription factor controlling protein expression in cell cycle arrest, DNA repair, apoptosis and senescence [53]. Mutation in p53 has been found to increase gain of function activities (p53^R273H^ and p53^R280K^) and encourage mutant binding to SREBPs, which of course leads to aberrant lipid metabolism, amplifying tumour progression thereby increasing fatty acid synthesis [53]. Conversely, TSG protein RB1 (a critical transcriptional corepressor in prevention of tumour development and progression) loss of function (LOF) has been associated with progression to castration-resistant PCa [54]. Furthermore, LOF of RB1 is associated with alteration in multiple metabolic pathways including lipid, amino acid and peptide metabolism; however, the specifics of lipid dysregulation are unknown [54]. More recently, Myc has also been associated with key FA synthesis genes including ACLY, ACC1 and FAS in PCa [55]. Hi-Myc transgenic mice with prostate-specific overexpression of Myc demonstrated increased circulating levels of total free FAs [55]. As previously noted, FA metabolism is intricately related to PCa development for ATP generation through β-oxidation supporting the excess energy requirements for growth and proliferation of cancer cells [24]. In light of these genetic alterations associated with more aggressive disease variants, recent studies have attempted to identify specific molecular features of aggressive PCa.

Aggressive variant PCa (AVPC) refers to AR-independent forms of PCa [56]. Clinically, AVPC is characterised by rapid disease progression including hormone refractory disease and visceral metastases. PTEN, p53, RB1 and Myc have been found to have more frequent alterations in AVPC and is characterised by a combination of alterations similar to small-cell PCa and therefore have direct clinical relevance with platinum-based treatment [56,57]. Hence, treatment of AVPC as similar to small-cell PCa has demonstrated improvement in progression-free survival (65.4% first-line and 33.8% second-line) and median overall survival (16 months (95% CI, 13.6–19.0 months)) after first line carboplatin and docetaxel and second line etoposide and cisplatin, respectively [58].

## 8. Novel Pharmacology Treatments

CYP11A is an essential enzyme in catalysing the initial step of steroid hormone biosynthesis. Inhibition of CYP11A has been hypothesised to halt the synthesis of all steroid hormones. A recent study by Karimaa et al. (2022), showed development of the first-in-class oral CYP11A1 inhibitor, ODM-208, which is non-steroidal and selective. Administration to animal models and 6 human patients with metastatic castration-resistant PCa demonstrated rapid, complete and reversible inhibition within a few weeks, in conjunction with steroid hormone replacement. The authors note that ODM-208 administration is feasible with concomitant corticosteroid replacement therapy which is an exciting potential avenue for treatment of castration-resistant PCa [59].

## 9. Conclusions

Prostate cancer is intrinsically related to lipid metabolism and androgens through several notable pathways. This review details the multi-level relationship between lipid metabolism and prostate cancer. The understanding of this interconnectivity is continually evolving and offers a clear pathway towards propagating novel therapeutic management of an ever-growing global cancer disease burden.

## Figures and Tables

**Figure 2 cancers-15-00341-f002:**
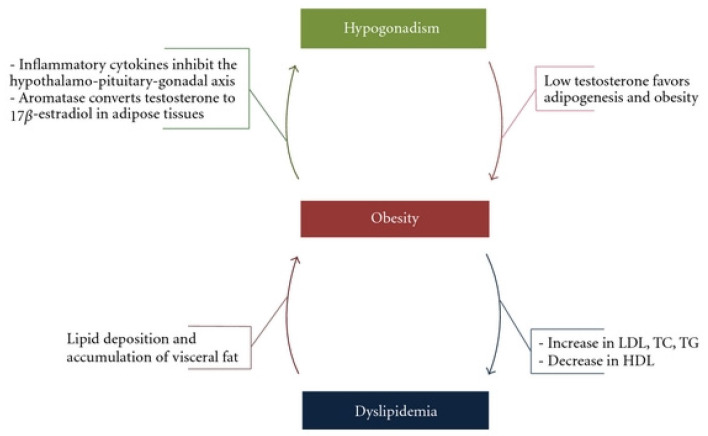
The vicious cycle of hypogonadism, obesity and dyslipidaemia by Fahed et al. via Creative Commons Attribution License [22].

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
