# Peer review of "Dysfunctional Lipid Metabolism—The Basis for How Genetic Abnormalities Express the Phenotype of Aggressive Prostate Cancer"

_cancers, 2023, doi:10.3390/cancers15020341_

Round 1

Reviewer 1 Report

My comments are given in the attached word file.

Reviewer 2 Report

The article discusses the relationship between lipid metabolism and prostate cancer (PCa) and how androgens and ADT (androgen deprivation therapy) can affect this relationship. The article also mentions potential mechanisms for dysregulated lipid metabolism in PCa, including amplification and overexpression of the pyruvate dehydrogenase complex (PDC) and co-deletion of promyelocytic leukemia (PML) and phosphatase and tensin homolog (PTEN). It is also mentioned that ADT can affect body composition by increasing body fat mass and decreasing lean mass, which may increase the risk of co-morbid conditions associated with metabolic syndrome. Overall, the information provided in the text appears to be scientifically sound and not misleading.

I suggest the authors to consider these points:

1. Along with PTEN, TP53, RB1, and MYC are well established drivers of aggressive prostate cancer. There is evidence to suggest that TP53, RB1, and MYC genetic alterations may be associated with dysfunctional lipid metabolism in cancer. The authors may provide further context in their articles:

Parrales A, Iwakuma T. p53 as a Regulator of Lipid Metabolism in Cancer. Int J Mol Sci. 2016;17(12):2074. Published 2016 Dec 10. doi:10.3390/ijms17122074

Mandigo AC, Yuan W, Xu K, et al. RB/E2F1 as a Master Regulator of Cancer Cell Metabolism in Advanced Disease. Cancer Discov. 2021;11(9):2334-2353. doi:10.1158/2159-8290.CD-20-1114

Zacksenhaus E, Shrestha M, Liu JC, et al. Mitochondrial OXPHOS Induced by RB1 Deficiency in Breast Cancer: Implications for Anabolic Metabolism, Stemness, and Metastasis. Trends Cancer. 2017;3(11):768-779. doi:10.1016/j.trecan.2017.09.002

Singh KB, Hahm ER, Kim SH, Wendell SG, Singh SV. A novel metabolic function of Myc in regulation of fatty acid synthesis in prostate cancer. Oncogene. 2021;40(3):592-602. doi:10.1038/s41388-020-01553-z

Dong, Y., Tu, R., Liu, H. et al. Regulation of cancer cell metabolism: oncogenic MYC in the driver’s seat. Sig Transduct Target Ther 5, 124 (2020). https://doi.org/10.1038/s41392-020-00235-2

2. Aggressive variants of prostate cancer (AVPC) has been defined by clinical factors, and the definition has evolved to by genetic and transcriptomic methods. The authors may cite those articles to introduce AVPCs and discuss the role of lipid metabolism in TP53, RB1, PTEN or MYC altered AVPCs.

Aparicio AM, Harzstark AL, Corn PG, et al. Platinum-based chemotherapy for variant castrate-resistant prostate cancer. Clin Cancer Res. 2013;19(13):3621-3630. doi:10.1158/1078-0432.CCR-12-3791

Aparicio AM, Shen L, Tapia EL, et al. Combined Tumor Suppressor Defects Characterize Clinically Defined Aggressive Variant Prostate Cancers. Clin Cancer Res. 2016;22(6):1520-1530. doi:10.1158/1078-0432.CCR-15-1259

Han H, Lee HH, Choi K, et al. Prostate epithelial genes define therapy-relevant prostate cancer molecular subtype. Prostate Cancer Prostatic Dis. 2021;24(4):1080-1092. doi:10.1038/s41391-021-00364-x

Round 2

Reviewer 2 Report

 appreciate the authors' efforts on this important topic in prostate cancer. Understanding lipid metabolism in prostate cancer could lead to new therapeutic opportunities, such as a healthy diet or regular exercise, or even the development of drugs that target lipid metabolism. I accept the revised article.